# Associations between Prediagnostic Circulating Bilirubin Levels and Risk of Gastrointestinal Cancers in the UK Biobank

**DOI:** 10.3390/cancers13112749

**Published:** 2021-06-01

**Authors:** Nazlisadat Seyed Khoei, Karl-Heinz Wagner, Robert Carreras-Torres, Marc J. Gunter, Neil Murphy, Heinz Freisling

**Affiliations:** 1Department of Nutritional Sciences, Faculty of Life Sciences, University of Vienna, Althanstrasse 1, 1090 Vienna, Austria; nazlisadat.seyedkhoei@univie.ac.at (N.S.K.); karl-heinz.wagner@univie.ac.at (K.-H.W.); 2Colorectal Cancer Group, ONCOBELL Program, Bellvitge Biomedical Research Institute (IDIBELL), Avinguda de la Granvia de l’Hospitalet 199-203, L’Hospitalet de Llobregat, 08908 Barcelona, Spain; rcarreras@idibell.cat; 3Nutrition and Metabolism Branch, International Agency for Research on Cancer (IARC-WHO), 150 Cours Albert Thomas, CEDEX 08, 69372 Lyon, France; gunterm@iarc.fr

**Keywords:** gastrointestinal cancers, cancer risk, bilirubin, UK Biobank

## Abstract

**Simple Summary:**

Evidence from experimental studies suggests that bilirubin, a metabolic by-product of hemoglobin breakdown, has anticancer activity and may, therefore, reduce the risk of gastrointestinal (GI) cancers. We conducted a prospective study among 440,948 participants in the UK Biobank and found that higher prediagnostic circulating bilirubin levels were robustly associated with a lower risk of developing esophageal adenocarcinoma, which is compatible with the antioxidant hypothesis of bilirubin. We further observed negative associations between bilirubin and risk of colorectal cancer, which were less robust and could be due to reverse causality, whereby undiagnosed cancer affects bilirubin levels. The observed positive associations between bilirubin and risk of hepatobiliary cancers may indicate underlying liver disease processes. No associations were found for cancers of the mouth, stomach, and pancreas. Bilirubin is a novel biomarker for disease development that is routinely measured in clinical settings. Provided that our findings are replicated in further studies, circulating bilirubin could serve as a future risk stratification marker for certain GI cancers.

**Abstract:**

We investigated associations between serum levels of bilirubin, an endogenous antioxidant, and gastrointestinal cancer risk. In the UK Biobank, prediagnostic serum levels of total bilirubin were measured in blood samples collected from 440,948 participants. In multivariable-adjusted Cox proportional hazard regression, we estimated hazard ratios (HR) and 95% confidence intervals (CI) for associations between bilirubin levels and gastrointestinal cancer risk (colorectum, esophagus, stomach, mouth, pancreas, and liver). After a median follow-up of 7.1 years (interquartile range: 1.4), 5033 incident gastrointestinal cancer cases were recorded. In multivariable-adjusted models, bilirubin levels were negatively associated with risk of esophageal adenocarcinoma (EAC, HR per 1-SD increment in log-total bilirubin levels 0.72, 95%CI 0.56–0.92, *p* = 0.01). Weak and less robust negative associations were observed for colorectal cancer (CRC, HR per 1-SD increment in log-total bilirubin levels 0.95, 95%CI 0.88–1.02, *p* = 0.14). Bilirubin levels were positively associated with risk of hepatocellular carcinoma (HCC, HR per 1-SD increment in log-total bilirubin levels 2.07, 95%CI 1.15–3.73, *p* = 0.02) and intrahepatic bile duct (IBD) cancer (HR per 1-SD increment 1.67, 95%CI 1.07–2.62, *p* = 0.03). We found no associations with risks of stomach, oral, and pancreatic cancers. Prediagnostic serum levels of bilirubin were negatively associated with risk of EAC and positively associated with HCC and IBD cancer. Further studies are warranted to replicate our findings for specific GI cancers.

## 1. Introduction

Gastrointestinal (GI) cancers are among the most common malignancies worldwide and contributed to approximately 26% of incident cancers and 35% of cancer-related deaths in 2018 [1]. With the possible exception of gastric non-cardia cancer, where incidence and mortality rates have been declining, incidence of esophageal adenocarcinoma (EAC), and cancers of the gastric cardia, colorectum, liver and pancreas are projected to increase globally. Primary prevention and early detection measures will be critical to reduce the global burden of these cancers [1].

The GI tract is highly exposed to reactive oxygen species (ROS) from external and internal sources, such as from cigarette smoke, alcohol consumption, viral/bacterial infections, or inflammatory disorders [2]. An imbalance between ROS and antioxidants can lead to oxidative-stress-induced damage to DNA, which is one of the major pathways that can lead to cancer development [3].

Antioxidants are compounds that can protect against oxidative stress, which is why the antioxidant prevention of GI cancers has drawn considerable attention [4]. Humans ingest exogenous antioxidants from foods, but also possess a variety of endogenous antioxidant compounds that may prevent oxidative damage [5]. Serum bilirubin, a byproduct of hemoglobin breakdown, has been shown to have potent endogenous antioxidant properties [6,7,8,9,10]. Bilirubin has an inhibitory impact on nicotinamide adenine dinucleotide phosphate (NADPH) oxidase activity, which explains much of the profound antioxidant property of heme oxygenase-1 (HO-1). HO-1 is the key enzyme in heme degradation to biliverdin, the latter being converted into bilirubin [11]. Congenital under-expression of hepatic uridine-diphosphoglucuronate glucuronosyltransferase1A1 *(UGT1A1)* causes mild chronic unconjugated hyperbilirubinemia, known as Gilbert’s syndrome (GS). Individuals with GS have mildly raised total bilirubin levels in the blood (≥17.1 µM/L) with normal serum activities of liver transaminases, biliary damage markers, and red blood cell counts [12]. Under physiologic conditions, total bilirubin is the summation of indirect (~80 to 85%) and direct (~15 to 20%) bilirubin [13]. The GS polymorphism is present in 10% of Europeans and 25% of individuals of African descent, although not all are exhibiting the phenotypic expression as overt hyperbilirubinemia (≥17.1 µM/L) [14,15,16].

Bilirubin may be particularly relevant in GI cancer development given that *UGT1A1* is highly expressed in GI tissues [17] and that the liver, bile ducts, colon and rectum, and pancreas are important organs in metabolizing and excreting bilirubin [18]. Findings from our previous in vitro studies supported anti-mutagenic effects of bilirubin, which may be particularly relevant for gut health. Intestinally cyclic tetrapyrroles, which are part of the bile pigment family, prevented genotoxicity induced by heterocyclic amines (HCAs) and resulted in apoptotic death in cancer cells [19,20,21,22]. However, the hypothesis that higher circulating bilirubin levels, which are indicative of bilirubin metabolism, are related to GI cancer risk is understudied. The few prospective studies investigating circulating bilirubin associated with colorectal cancer (CRC) and hepatocellular carcinoma (HCC) reported inconsistent results [13,23,24]. The genetically predicted high activity of UGT1A1, indicative of lower levels of circulating bilirubin, was positively associated with risk of esophageal squamous cell carcinoma (ESCC) [25], and not associated with risk of oral cancer [26]. Jiraskova et al. [27] reported a negative association between the *UGT1A1**28 polymorphism (presence of 7 TA repeats in the promoter region of the gene) and CRC risk in men; however, a positive association was reported in a Macedonian retrospective case-control study in men [28]. We are not aware of other studies investigating circulating levels of bilirubin associated with risks of esophageal, stomach, oral or pancreatic cancers.

We investigated associations between pre-diagnostic circulating total bilirubin levels and risk of GI cancers (colorectum, esophagus, stomach, mouth, pancreas, and liver) in the UK Biobank (UKB) study, a large prospective cohort including >500,000 participants.

## 2. Methods

### 2.1. Study Population

The UKB is a large, population-based prospective cohort study which is designed to provide data on the lifestyle, genetic, and environmental factors of important diseases [29]. This research has been conducted using the UKB Resource under application number 25897. Between 2006 and 2010, over 500,000 individuals (54% women) aged 40–69 years were recruited from one of 22 study centers across the UK (England, Wales, and Scotland). Approximately 9.2 million people were invited by the UKB to participate in this study through postal invitation with a telephone follow-up (response rate: 5.7%). All participants were registered with the UK National Health Service (NHS) and lived within 40 km of one of those 22 centers. The UKB has approval from the North West Multi-centre Research Ethics Committee, the National Information Governance Board for Health and Social Care in England and Wales, and the Community Health Index Advisory Group in Scotland. In addition, an independent Ethics and Governance Council was formed in 2004 to oversee UK Biobank’s continuous adherence to the Ethics and Governance Framework that was developed for the study (http://www.ukbiobank.ac.uk/ethics/, accessed on 3 July 2019). All participants provided written informed consent.

At the baseline assessment, participants completed a touchscreen self-reported questionnaire providing information on socio-demographics (age, sex, education, and Townsend deprivation index), health and medical history (vasectomy, hypertension, and diabetes), and lifestyle exposures (including smoking status, diet, physical activity, and alcohol consumption). Participants underwent anthropometric measurements, including body weight, height, and waist and hip circumference. Blood samples were collected, labeled, centrifuged, and stored at −80°C from all participants at recruitment and also from a subset of ~20,000 participants who re-attended the study center for a repeat assessment visit between 2012 and 2013.

Exclusions prior to the onset of analyses were participants with prevalent cancer at recruitment (*n* = 27,264) and participants without a total bilirubin measurement (*n* = 31,373). The final analytical cohort consisted of 440,948 participants.

### 2.2. Blood Collection and Laboratory Methods

As part of the UKB Biomarker Project, serum levels of total bilirubin and direct bilirubin were determined by a colorimetric assay (Beckman Coulter United Kingdom Ltd., Beckman Coulter AU5800 analyzer). Information on assay performance and the Coefficients of Variation (CVs) have been published [30]. The average within-laboratory CV for low, medium, and high internal quality control level samples for total and direct bilirubin ranged from 1.48 to 1.92% and 1.73 to 2.60%, respectively. A total of 15,611 participants had total bilirubin levels measured in blood samples collected at both the recruitment and repeat assessment visit (median of 4 years apart).

### 2.3. Follow-Up and Outcomes

Incident cancer cases and cancer cases recorded first in death certificates within the UKB cohort were identified through linkage to national cancer and death registries. Prevalent cancer cases were identified by linkage to cancer registries. Participants were followed from their baseline visit until the end of October 2015 for Scotland and the end of March 2016 for England and Wales.

Cancer incidence data were coded using the 10th Revision of the International Classification of Diseases (ICD-10). Separate analyses were conducted for colorectal (C18–C20), esophageal (C15), oral (C04), stomach (C16), liver (C22), and pancreatic (C25) cancers.

CRC cases were classified by location (anatomical sub-site), as cancers of colon (C18) and rectum (C19–C20). Cases of esophageal cancer were classified by histology, and these topography codes were combined with ICD-3 for oncology morphological codes [31] to further categorize esophageal cancer into adenocarcinoma (8140, 8144, 8145, 8260, 8480, 8481, 8490, and 8574) and ESCC (8070 and 8071). Stomach cancer was classified by location (anatomical sub-site), where possible, as cancers of the stomach cardia (C16.0) and stomach non-cardia site (C16.1–16.5). Liver cancer cases were classified as HCC (C22.0) and intrahepatic bile duct (IBD, C22.1).

### 2.4. Statistical Analysis

To assess reproducibility between the two measurements of total bilirubin available in a subsample of participants, we calculated intra-class correlation coefficients (ICC) by dividing the between-person variance by the sum of the between- and within-person variances. Given that the two measurements of total bilirubin were a median of 4 years apart, the ICC reflects both technical and biological within-person variation.

Cox proportional hazards regression was used to estimate cause-specific hazard ratios (HRs) and 95% confidence intervals (CIs) for associations between total bilirubin levels and incident cancers of the colorectum (colon and rectum), esophagus (EAC and ESCC), stomach (cardia and non-cardia), mouth, pancreas, and liver (HCC and IBD). The time variable in all models was age. Entry time was age at recruitment and exit time was age at whichever of the following came first: cancer diagnosis, death, or the last date at which follow-up was completed. All models were stratified by sex (except when models were run separately in men and women), age at recruitment in 5-year categories (<45, 45–49.9, 50–54.9, 55–59.9, 60–64.9, and ≥ 65 years), socio-economic status (Townsend deprivation index quintiles), and the recruitment centers [32,33].

Total bilirubin was modelled on a continuous scale (per 1-SD increment of log-total bilirubin levels) for men and women combined and by sex (except when number of events for a given cancer site was limited).

Based on prior knowledge and to avoid over-adjustment, we decided to control for confounding for each cancer separately by considering risk factors with strong evidence for each individual cancer. The multivariable models were adjusted for waist circumference (per 5 cm), smoking status and intensity (never, former, current- < 15/day, current- ≥ 15/day, current-intensity unknown, unknown), alcohol consumption frequency (never, special occasions only, 1–3 times/month, 1–2 times/week, 3–4 times/week, daily/almost daily, unknown), and qualifications (Certificates of secondary education/Ordinary-levels/General Certificates of Secondary Education or equivalent, National Vocational Qualification/Higher National Diploma/Higher National Certificate/Advanced-levels/Advanced Subsidiary-levels or equivalent, other professional qualifications, college/university degree, none of the above) [32,33].

The HCC and IBD cancer models were further adjusted for total physical activity (<10, 10–19.9, 20–39.9, 40–59.9, ≥ 60 metabolic equivalent hours/week), and height (per 10 cm). The pancreatic cancer model was further adjusted for prevalent diabetes [32], and the CRC model was additionally adjusted for frequency of red/processed meat intake (<2, 2–2.99, 3–3.99, ≥ 4 occasions/week), family history of CRC (no/yes), total intake of fruits and vegetables (tablespoons/day), regular aspirin/ibuprofen use (no/yes), and if participants had ever had menopausal hormone therapy (no/yes) [32,33].

Analyses were conducted by sex and anatomical sub-site/histology. The heterogeneity of associations by sex and across subsites was assessed by calculating log-likelihood ratio (LR). We also investigated potential non-linear dose–response associations between circulating levels of total bilirubin and GI cancer risk by applying cubic spline models combined with an LR test.

The associations between circulating total bilirubin and GI cancers were further assessed across subgroups of sex, median body mass indenx (BMI, 26.7 kg/m^2^), age at recruitment (50 years), and smoking status (never, former, and current). For these subgroup analyses, we used a more stringent *p*-value < 0.002 to correct for multiple testing (0.05 divided by the number of tests [*n* = 24]). In contrast, we did not adjust our main analysis for the six cancer outcomes for multiple testing given that our hypothesis was based on a strong prior hypothesis. In a sensitivity analyses, we excluded those participants with less than 2 years of follow-up to assess potential reverse causation. We report results for total bilirubin after adjusting for regression dilution by dividing the respective HRs (and 95%CIs) by the regression dilution ratio (RDR) value (=0.72). We also report results for direct and indirect bilirubin for the fully adjusted models. Last, we assessed the influence of high levels of circulating liver enzymes on our results by excluding participants in the highest decile of circulating alanine transaminase (ALT), aspartate transaminase (AST), alkaline phosphatase (ALP), and gamma-glutamyl transpeptidase (GGT).

Analyses were conducted in Stata version 15.0 (Stata Corp, College Station, TX, USA). Statistical tests were all two-sided and a *p*-value < 0.05 was considered statistically significant.

## 3. Results

### 3.1. General Characteristics of the Study Population

During a median follow-up time of 7.1 years (interquartile range: 1.4), incident cases of the following GI cancers were recorded: 3002 colorectum (1989 colon and 1013 rectum), 338 esophageal adenocarcinoma, 124 esophageal squamous cell, 139 stomach cardia, 92 stomach non-cardia, 524 mouth, 559 pancreas, 135 hepatocellular, and 120 intrahepatic bile duct.

The characteristics of the study population by tertiles of total bilirubin levels are summarized in Table 1. Compared to the lowest tertile of total bilirubin, participants in the highest tertile had on average a lower waist circumference, were less likely to be current smokers, were less likely to have diabetes, and were more likely to have a college/university degree.

The distribution of bilirubin levels was positively skewed with geometric means of 9.51 µM/L (95%CI 9.49–9.52) for men and 7.56 µM/L (95%CI 7.55–7.58) for women, which were within the clinical reference range of bilirubin levels (5–17 µM/L).

The reproducibility (ICC) of total bilirubin levels measured at both the recruitment and repeat assessment visit (*n* = 15,611 participants; median of 4 years apart) was 0.72 (95%CI 0.71–0.73) for both sexes.

### 3.2. Associations between Total Bilirubin Levels and Risk of GI Cancers

In the multivariable model, circulating total bilirubin levels were negatively associated with risk of CRC (HR per 1-SD increment in log-total bilirubin levels 0.95, 95%CI 0.88–1.02, *p* = 0.14) (Table 2), with similar associations found for men and women (P-heterogeneity = 0.48). Evidence of non-linearity (U-shaped) of the association between total bilirubin levels and CRC was found (*p* = 0.02) (Figure 1A). The dose–response analysis using restricted cubic splines indicated a threshold level of circulating bilirubin of approximately 10 µM/L, after which the negative association plateaued (Figure 1A).

Circulating total bilirubin levels were negatively associated with EAC risk (HR per 1-SD increment in log-total bilirubin levels 0.72, 95%CI 0.56–0.92, *p* = 0.01) (Table 2), with similar associations found for men and women (P-heterogeneity = 0.55). Evidence of non-linearity (U-shaped) of the association between total bilirubin levels and EAC was found (*p* = 0.06) (Figure 1B). The dose–response analysis using restricted cubic splines showed a shape of association similar to CRC with an observed lowest risk between 10 and 13 µM/L of total bilirubin (Figure 1B). Total bilirubin levels were also negatively related to ESCC risk, although this association did not reach the threshold of statistical significance (Table 2). We found little evidence that total bilirubin levels were associated with risks of stomach cardia, stomach non-cardia, and oral cancers. There was suggestive evidence that associations between total bilirubin levels and risk of pancreatic cancer differed by sex (P-heterogeneity = 0.05), with HRs equal to 0.83 (95%CI 0.65–1.04) and 1.11 (0.89–1.38) for men and women, respectively, although these associations did not reach the threshold of statistical significance. Total bilirubin levels were positively associated with risks of HCC (HR per 1-SD increment in circulating log-total bilirubin levels, 2.07, 95%CI 1.15–3.73) and IBD cancer (HR per 1-SD increment in circulating log-total bilirubin levels, 1.67, 95%CI 1.07–2.62) in the continuous models. There was evidence of non-linearity for HCC, which, however, was likely due to the limited number of events in the lower or higher distributions of total bilirubin.

### 3.3. Subgroup and Sensitivity Analyses

There was little evidence of heterogeneity for the associations between total bilirubin levels and risk of GI cancers according to subgroups of other risk factors with a few exceptions. For ESCC (P-heterogeneity = 0.01) and pancreatic cancer (P-heterogeneity = 0.008), we detected heterogeneity according to smoking status, with negative associations, albeit not statistically significant, only found for never smokers (Appendix A).

We also found evidence of heterogeneity of the total bilirubin and IBD cancer risk according to BMI (P-heterogeneity = 0.0004), with a positive association only found for the above median BMI group. Results were generally similar when GI cancer cases occurring during the first two years of follow-up were excluded, except for CRC, where the association was attenuated and became null (HR per 1-SD increment in circulating log-total bilirubin levels, 1.03, 95%CI 0.95–1.13) (Table 3).

The results were similar after adjusting the HR (and 95%CI) for regression dilution using the RDR (Appendix A) and for direct and indirect bilirubin (Appendix A). Excluding participants (*n* = 560) in the highest decile of circulating levels of ALT, AST, ALP, and GGT, did not alter our results (Appendix A).

## 4. Discussion

In this large-scale prospective study, higher pre-diagnostic circulating levels of total bilirubin were strongly negatively associated with risk of EAC. This finding was consistent for men and women and after stratification by BMI and smoking status. We found positive associations between circulating total bilirubin levels and risks of HCC and IBD cancer. For CRC, the negative association found in our main analysis was likely a consequence of reverse causality. Nevertheless, circulating bilirubin could be useful for detecting the presence of subclinical disease.

Oxidative-stress-induced damage to DNA is accepted as a major contributing factor to carcinogenesis [34]. As bilirubin is a potent antioxidant, it may protect against malignancy [35]; however, the exact mechanisms of the anticarcinogenic effects of bilirubin are still not completely understood. Experimental data [36,37,38] support a role for bilirubin and its downstream signaling pathways in tumorigenesis. Ollinger et al. have found that bilirubin can induce cell cycle arrest and apoptosis in abnormally proliferating cells and contributes to the defense against cancer by interfering with procarcinogenic signaling pathways [37].

We are not aware of other prospective epidemiological studies investigating the role of circulating levels of bilirubin in esophageal cancer development. Dura et al. [25] found in a case-control study with esophageal cancer patients (*n* cases = 351 Caucasians) that genetically predicted high activity of UGT1A1, which is predictive of low serum levels of bilirubin, was associated with an increased risk of ESCC. However, since ESCC is rare in the Netherlands, they could not firmly establish their findings. A direct protective effect of the UGT1A1 enzyme may not be present in the esophagus, as this enzyme is not highly expressed in esophageal cells, whereas a systemic effect of circulating bilirubin could be exerted [25,39]. The results of our study are compatible with this hypothesis and support further studies on the utility of circulating bilirubin as a routinely collected biomarker for EAC risk-stratification.

The positive associations found in our analyses for total bilirubin and HCC and IBD cancer risk were consistent with a prior nested case-control study in the European Prospective Investigation into Cancer and nutrition (EPIC) cohort [24]. Abnormally high bilirubin levels in patients with HCC were also associated (≥25.7 μM/L) with cancer aggressiveness [40]. As bilirubin is conjugated by UGT1A1 in the liver, liver dysfunction may result in higher circulating bilirubin levels due to some underlying liver disease process [41] and correlates with the severity of the illness [42,43].

For CRC, the negative association in our main analysis was attenuated when participants with less than 2 years of follow-up were excluded, which is suggestive of reverse causation, whereby subclinical diseases also affect circulating bilirubin levels. In our recent case-control study, nested in the prospective EPIC cohort [13], we found that higher circulating levels of unconjugated bilirubin, the main component of total bilirubin, were positively associated with CRC risk in men and negatively associated in women [13]. Possible explanations for the inconsistent results found between EPIC and the current study are unclear. Some of this heterogeneity may be explained by differences in study designs, limited sample size in previous studies, and measurements of total bilirubin levels or its components [13,23,27,44,45]. However, there may also be heterogeneity related to population sub-group characteristics, especially by sex and *UGT1A1* genotype, which suggests that further refined analyses are needed.

We found little evidence that total bilirubin was associated with oral cancer risk. Lacko et al. [26] found no association between the *UGT1A1**28 polymorphism and risk of the oral cavity. Cancer causing types of human papilloma virus (HPV) is the leading cause of oropharyngeal cancers. To our knowledge, there is no evidence that the antioxidative/anticarcinogenic role of bilirubin can affect the carcinogenic action of HPV. The expression of *UGT1A1* is low in the oropharynx, which precludes a direct protective role of this enzyme; however, an indirect effect of circulating bilirubin cannot be excluded [37].

To our knowledge, this is the first prospective study investigating the association between circulating bilirubin levels with risks of stomach and pancreatic cancer and, overall, we found little evidence of associations. However, the suggestive evidence (p-heterogeneity = 0.05 did not reach our set level of significance = 0.002) for differences among men and women deserves further investigation. Sex hormones, which are known to influence UGT1A1 activity [46], and differences in *UGT1A1* expression between men and women, leading to differential circulating levels [47], might partly explain the sex differences in pancreatic cancer risk. We previously reported suggestive evidence of differences between men and women related to risk of pancreatic cancer [48] and CRC [13] using a genetic instrument approach, yet opposite in direction (suggestive higher risk in men and lower risk in women) compared to the present analysis. Our sex-stratified findings for pancreatic cancer could therefore also be due to chance.

This was the most comprehensive study to examine the associations between circulating bilirubin levels and GI cancer risk. This allowed us to directly compare associations across different cancer types with uniform confounder adjustments, and to suggest a range of total bilirubin levels associated with the lowest risk of developing certain types of cancer (i.e., 10–13 μM/L, EAC and CRC). These are novel and potentially important thresholds regarding cancer risk, but replication with a larger number of events is needed. Uniquely, our study followed a full cohort, rather than a nested case-control study design, as bilirubin levels were measured in all UKB participants. The availability of repeat total bilirubin measurements in a subset of participants (*n* = 15,611) meant we were able to correct our HRs for regression dilution bias, thereby diminishing the effects of measurement error and within-person variability. A limitation of our study was that for most participants total bilirubin levels were measured once at baseline. However, our reproducibility analysis using the repeated bilirubin measures from a subset of participants collected over a median of 4 years apart found an ICC value of 0.72, demonstrating that a single measure of bilirubin levels provides a fairly good estimate of longer-term exposure. A further limitation was the low number of events for some cancer sites, which resulted in relatively wide confidence intervals, particularly for hepatocellular carcinoma (N cases = 135), intrahepatic bile duct (N cases = 120), and to some extent also EAC (N cases = 338).

## 5. Conclusions

In this comprehensive analysis of circulating total bilirubin levels and GI cancer risk, we found a novel and robust negative association for EAC risk. If this association is validated in subsequent studies, circulating bilirubin measurements could be used to risk stratify people at higher risk of developing EAC. We further observed a weak and less robust negative association between total bilirubin levels and CRC risk and positive associations with risks of HCC and IBD.

## Figures and Tables

**Figure 1 cancers-13-02749-f001:**
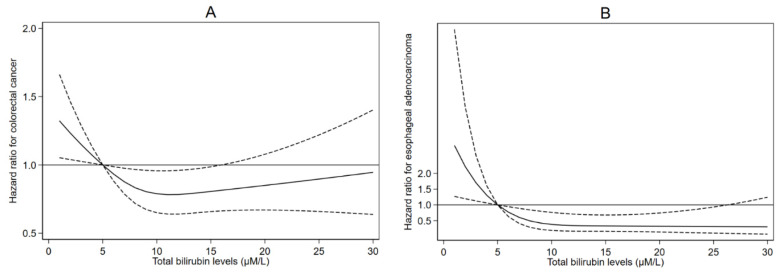
Cubic spline modeling of bilirubin levels in relation to cancer risk. This graph shows multivariable-adjusted cox proportional hazard regression (solid line) with 95% confidence intervals (dashed lines) for the association between total bilirubin levels (µM/L) and the incident colorectal cancer (**A**), and esophageal adenocarcinoma (**B**) from the UK Biobank. It was modeled by restricted cubic splines with 3 knots at percentiles 10th, 50th, and 90th in a regression model to evaluate the linearity hypothesis.

**Table 1 cancers-13-02749-t001:** Characteristics of UK Biobank study participants by category of circulating total bilirubin levels (*N* = 440,948).

	**Log-Total Bilirubin Tertile (μM/L)**					
**Baseline Characteristic**	**Men**			**Women**		
	**1 (1.43–7.94)**	**2 (7.95–10.64)**	**3 (10.65–144.52)**	**1 (1.08–6.36)**	**2 (6.37–8.41)**	**3 (8.42–115.75)**
**Cases (N)**						
Colorectal cancer	614	555	562	462	428	381
Colon cancer	370	352	336	342	302	287
Rectal cancer	244	203	226	120	126	94
Esophageal adenocarcinoma	124	86	81	21	16	10
Esophageal squamous-cell carcinoma	31	15	13	24	21	20
Stomach cardia cancer	41	40	32	10	11	5
Stomach non-cardia cancer	18	22	16	16	11	9
Oral cancer	126	111	110	60	60	57
Pancreatic cancer	110	118	77	76	98	80
Hepatocellular carcinoma	15	35	60	3	7	15
Intrahepatic bile duct cancer	17	14	23	23	18	25
Age at recruitment (years) °	56.3 (8.1)	56.7 (8.1)	56.6 (8.3)	56.1 (7.9)	56.6 (7.9)	55.7 (8.2)
Body mass index (kg/m^2^) °	28.2 (4.5)	27.8 (4.1)	27.5 (4.0)	27.9 (5.4)	27.0 (5.1)	26.3 (4.9)
Waist circumference (cm) °	97.9 (11.8)	96.8 (11.1)	95.9 (10.9)	86.7 (12.9)	84.4 (12.3)	82.7 (12.0)
Height (cm) °	174.9 (6.8)	175.8 (6.8)	176.2 (6.9)	161.8 (6.3)	162.5 (6.3)	163.1 (6.3)
Smoking status (%)						
Never	30	34	36	32	33	34
Former	33	34	33	32	34	34
Current	49	29	21	46	31	23
Alcohol consumption status (%)						
Never	39	30	31	40	32	28
Former	41	30	29	43	31	26
Current	33	34	34	33	34	34
Physical activity (MET§ hour/week) (%)						
<10	36	33	31	36	33	31
>60	34	33	33	31	34	35
Education (%)						
College/university degree	31	33	36	30	33	36
Family history of CRC (%)						
Yes	33	34	34	33	34	33
Prevalent diabetes (%)						
Yes	41	30	29	42	30	28
Aspirin/Ibuprofen Use (%)						
Yes	34	33	33	35	33	32
Red and processed meat (%)						
<2/week	31	34	35	32	33	34
≥4/week	35	33	32	35	33	32
Ever use of hormone therapy (%) ¤						
Yes				35	34	31

° Mean and standard deviation. § MET = metabolic equivalents. ¤ Only among women.

**Table 2 cancers-13-02749-t002:** Risk (hazard ratios) of gastrointestinal cancers associated with circulating total bilirubin levels in the UK Biobank.

	Both Sexes	Men	Women
Colorectal Cancer (CRC)			
N cases	3002	1731	1271
Crude HR (95%CI) *	0.93 (0.87–1.00)	0.92 (0.85–1.01)	0.96 (0.86–1.06)
*p*	0.05	0.07	0.39
HR (95%CI) **	0.95 (0.88–1.02)	0.93 (0.85–1.02)	0.98 (0.88–1.09)
*p*	0.14	0.11	0.74
Pnon-linearity	0.02	N/A	N/A
Colon Cancer			
N cases	1989	1058	931
Crude HR (95%CI) *	0.93 (0.85–1.01)	0.91 (081–1.01)	0.96 (0.85–1.08)
*p*	0.08	0.08	0.49
HR (95%CI) **	0.95 (0.87–1.04)	0.92 (0.82–1.03)	1.00 (0.88–1.13)
*p*	0.26	0.15	>0.9
Pnon-linearity	0.07	N/A	N/A
Rectal Cancer			
N cases	1013	673	340
Crude HR (95%CI) *	0.95 (0.84–1.07)	0.95 (0.82–1.09)	0.95 (0.78–1.16)
*p*	0.38	0.48	0.61
HR (95%CI) **	0.94 (0.83–1.07)	0.93 (0.80–1.09)	0.97 (0.77–1.21)
*p*	0.36	0.37	0.76
Pnon-linearity	0.29	N/A	N/A
Esophageal Adenocarcinoma (EAC)			
N cases	338	291	47
Crude HR (95%CI) *	0.69 (0.55–0.87)	0.72 (0.56–0.91)	0.60 (0.33–1.10)
*p*	0.002	0.01	0.10
HR (95%CI) **	0.72 (0.56–0.92)	0.74 (0.56–0.97)	0.84 (0.40–1.78)
*p*	0.01	0.03	0.65
Pnon-linearity	0.06	N/A	N/A
Esophageal Squamous-Cell Carcinoma (ESCC)			
N cases	124	59	65
Crude HR (95%CI) *	0.82 (0.55–1.21)	0.77 (0.40–1.48)	0.86 (0.55–1.35)
*p*	0.32	0.43	0.51
HR (95%CI) **	0.74 (0.44–1.24)	N/A	0.68 (0.33–1.39)
*p*	0.25	N/A	0.29
Pnon-linearity	N/A	N/A	0.11
Stomach Cardia Cancer			
N cases	139	113	26
Crude HR (95%CI) *	1.01 (0.71–1.45)	1.14 (0.77–1.67)	0.53 (0.20–1.43)
*p*	0.07	0.51	0.21
HR (95%CI) **	1.33 (0.84–2.11)	1.59 (0.90–2.82)	N/A
*p*	0.23	0.11	N/A
Pnon-linearity	0.86	N/A	N/A
Stomach Non-Cardia Cancer			
N cases	92	56	36
Crude HR (95%CI) *	0.67 (0.36–1.26)	0.83 (0.37–1.87)	0.56 (0.24–1.33)
*p*	0.22	0.66	0.19
HR (95%CI) **	0.88 (0.41–1.91)	3.23 (0.44–23.61)	0.15 (0.00–7.60)
*p*	0.74	0.25	0.34
Pnon-linearity	N/A	N/A	N/A
Oral Cancer			
N cases	524	347	177
Crude HR (95%CI) *	0.89 (0.75–1.05)	0.85 (0.69–1.06)	0.96 (0.74–1.24)
*p*	0.17	0.14	0.75
HR (95%CI) **	0.93 (0.77–1.13)	0.94 (0.73–1.20)	1.01 (0.75–1.35)
*p*	0.46	0.61	>0.9
Pnon-linearity	>0.9	N/A	N/A
Pancreatic Cancer			
N cases	559	305	254
Crude HR (95%CI) *	0.94 (0.81–1.09)	0.87 (0.70–1.06)	1.03 (0.84–1.25)
*p*	0.40	0.16	0.78
HR (95%CI) **	0.96 (0.82–1.13)	0.83 (0.65–1.04)	1.11 (0.89–1.38)
*p*	0.62	0.11	0.36
Pnon-linearity	N/A	N/A	N/A
Hepatocellular Carcinoma (HCC)			
N cases	135	110	25
Crude HR (95%CI) *	1.65 (1.21–2.49)	1.57 (1.08–2.29)	1.69 (1.03–2.79)
*p*	0.002	0.02	0.04
HR (95%CI) **	2.07 (1.15–3.73)	1.54 (0.62–3.27)	N/A
*p*	0.02	0.26	N/A
Pnon-linearity	0.02	N/A	N/A
Intrahepatic Bile Duct (IBD) Cancer			
N cases	120	54	66
Crude HR (95%CI) *	1.37 (0.69–1.96)	1.42 (0.86–2.36)	1.29 (0.81–2.06)
*p*	0.08	0.17	0.29
HR (95%CI) **	1.67 (1.07–2.62)	5.59 (0.88–35.75)	1.40 (0.78–2.50)
*p*	0.03	0.07	0.25
Pnon-linearity	>0.9	N/A	N/A

HR: hazard ratio, CI: confidence interval, N/A: not available or not assessed. All HRs are per 1-SD increment in circulating log-total bilirubin levels. * The crude multivariable cox regression model stratified by sex, Townsend deprivation index (quintiles), region of the recruitment assessment center, and age at recruitment. ** The multivariable models were adjusted for waist circumference (per 5 cm), smoking status and intensity (never, former, current- < 15/day, current- ≥ 15/day, current- intensity unknown, unknown), alcohol consumption frequency (never, special occasions only, 1–3 times/month, 1–2 times/week, 3–4 times/week, daily/almost daily, unknown), and qualification (Certificates of secondary education/Ordinary-levels/General Certificates of Secondary Education or equivalent, National Vocational Qualification/Higher National Diploma/Higher National Certificate/Advanced-levels/Advanced Subsidiary-levels or equivalent, other professional qualifications, college/university degree, none of the above). The HCC and IBD cancer models were further adjusted for total physical activity (<10, 10–19.9, 20–39.9, 40–59.9, ≥60 metabolic equivalent hours/week), and height (per 10 cm). The pancreatic cancer model was further adjusted for prevalent diabetes [32] and the CRC model was additionally adjusted for frequency of red and processed meat consumption (<2, 2–2.99, 3–3.99, ≥4 occasions/week), family history of CRC (no/yes), total intake of fruit and vegetable (tablespoons/day), regular aspirin/ibuprofen use (no/yes), and ever use of menopausal hormone therapy (no/yes).

**Table 3 cancers-13-02749-t003:** Excluding participants with less than 2 years of follow-up

Cancer Types	HR (95%CI)	*p*
Colorectal cancer (CRC)		
Both sexes	1.03 (0.95–1.13)	0.46
Men	0.98 (0.88–1.10)	0.73
Women	1.13 (0.99–1.29)	0.07
Colon cancer		
Both sexes	1.07 (0.96–1.20)	0.20
Men	0.98 (0.85–1.14)	0.80
Women	1.21 (1.03–1.42)	0.02
Rectal cancer		
Both sexes	1.00 (0.85–1.17)	>0.90
Men	1.02 (0.84–1.24)	0.83
Women	1.05 (0.79–1.40)	0.73
Esophageal adenocarcinoma (EAC)		
Both sexes	0.76 (0.56–1.03)	0.07
Men	0.76 (0.53–1.08)	0.13
Women	1.58 (0.55–4.53)	0.40
Esophageal squamous-cell carcinoma (ESCC)		
Both sexes	0.50 (0.22–1.13)	0.10
Stomach cardia cancer		
Both sexes	1.45 (0.77–2.73)	0.25
Stomach non-cardia cancer		
Both sexes	0.45 (0.15–1.36)	0.16
Oral cancer		
Both sexes	0.95 (0.75–1.21)	0.68
Men	0.92 (0.67–1.26)	0.61
Women	0.97 (0.65–1.47)	0.90
Pancreatic cancer		
Both sexes	0.97 (0.80–1.16)	0.72
Men	0.77 (0.58–1.02)	0.07
Women	1.19 (0.92–1.54)	0.19
Hepatocellular carcinoma (HCC)		
Both sexes	2.01 (1.10–3.68)	0.02
Intrahepatic bile duct (IBD) cancer		
Both sexes	1.36 (0.81–2.30)	0.30

HR: hazard ratio, CI: confidence interval. Multivariable-adjusted Cox proportional hazard regression models (HR per 1-SD increment in circulating log-total bilirubin levels), *N* = 431,500.

## Data Availability

The UK Biobank resource is available to bona fide researchers for health-related research in the public interest. All researchers who wish to access the research resource must register with UK Biobank by completing the registration form in the Access Management System (AMS—https://bbams.ndph.ox.ac.uk/ams/, accessed on 3 July 2019). Analytic codes will be made available to other researchers upon request.

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
