# Peer review of "Associations between Prediagnostic Circulating Bilirubin Levels and Risk of Gastrointestinal Cancers in the UK Biobank"

_cancers, 2021, doi:10.3390/cancers13112749_

Round 1

Reviewer 1 Report

Authors present a prospective study among 440,948 partecipants in the UK biobank looking for bilirubin levels andrisk of gastrointestinal cancer.

The sample is very large, the utility of the information is doubtful.

  1. What about direct and indirect bilirubin?
  2. Correlation between liver disease with high bilirubin levels and risk of hepatobiliary cancer is already known (HCC and cirrhosis, PSC and cholangiocarcinoma)
  3. I do not see usefulness in considering low total bilirubin levels for  colorectal cancer screening.

Reviewer 2 Report

Participants with prevalent cancer at recruitment were excluded. What is the rationale for this, is it because cancer modifies bilirubin levels? How was ‘prevalent cancer’ defined? What about participants with cancer diagnosed a long time before baseline (e.g. cured)? What about other conditions that may modify bilirubin levels?

Authors state they assessed the reproducibility of bilirubin measures by calculating ICCs, but if I understood correctly, the 2 measures were made a median 4 years apart, which leaves room for reasons other than technical for the measures to vary.

Regression dilution ratios assume that correcting measurement error would change the relative risk estimates fully in the direction of the non-corrected estimates, but this may not be the case if measurement error is e.g. non-differential. Here the measures were made a median 4 years apart and ICCs are ~0.70 (i.e. there might be virtually no measurement error) so it seems very inappropriate to use them to correct the RR estimates…

Models are adjusted for several relevant variables but there is still room for confounding, which should be acknowledged or further controlled where possible. In fact, the adjustment strategy is somewhat strange as most variables mentioned are associated with risk of any gastrointestinal cancer to some extent. Given none of these variables can be mediators, it might be best to use one comprehensive adjustment set for all cancer types.

Regarding excluding 2 years of follow-up for control of reverse causality, is there evidence that bilirubin levels are elevated in people with (pre-) cancer (other than pancreatic cancer)? (same comment as #1)

It seems quite a stretch to calculate a “P-trend” using the tertiles; any trend should be captured by the linear per SD analysis, shouldn’t it? In fact there is quite a bit of cherry picking in the results, for example those reported in the abstract seem to be the “most significant”, i.e. either tertiles or continuous per SD.

It is poor practice to categorise variables as the authors did, with clearly no justification for it, either biological, clinical or statistical... I suggest the authors take out all tertile analyses from the manuscript.

Finally, given the numbers of tests performed it is not surprising that a few are significant, e.g. heterogeneity tests. There seems to be an overall problem in the interpretation of the results; my understanding is that the authors found a potential negative (not “inverse” please correct) association with risk of oesophageal adenocarcinoma, and a potential positive association with HCC and IBD risk, both with wide confidence intervals, particularly for the latter. The rest of findings is very speculative and does not seem to be interpretable given sample size, except for colorectal cancer risk (for which there is, at best, a very weak association, that is furthermore maybe confounded or due to existing disease).

Reviewer 3 Report

While the manuscript is well-written, it is unclear why the authors decided to study association of circulating bilirubin levels with GI cancer risk.

Suggestions:

  1. Please expand on rationale on why circulating bilirubin levels were chosen to study in association to GI cancer risk.
  2. Include some more details on anti-oxidative properties of bilirubin. Connect it to GI cancers. Expand on why and how would bilirubin work in the GI setting - hypothesize about it.
  3. When referring to Gilbert's syndrome (GS) - comment do those patients have higher or lower incidence of GI cancers. Could you study population of GS patients?
  4. Since later in Discussion you comment on weaker inverse association of circulating bilirubin and CRC, please state this in the abstract as well.
  5. Please comment on difference in association of bilirubin and pancreatic cancer based on sex. (lines 247-251)
  6. Comment on how you envision the use of circulating bilirubin levels in clinic for early diagnostics of GI cancers? What threshold you would use (for each disease where you detected association)? What would be next course of action for those subjects?

Round 2

Reviewer 1 Report

The authors have sufficiently addressed the critical points

Reviewer 2 Report

The manuscript has been improved.

Reviewer 3 Report

I would like to thank the authors for responding to all raised questions/suggestions.